# The impact of anthropogenic inputs on lithium content in river and tap water

Hye-Bin Choi [1,2], Jong-Sik Ryu [3]*, Woo-Jin Shin [1] & Nathalie Vigier[4]

The use of lithium (Li) has dramatically increased during the last two decades due to the proliferation of mobile electronic devices and the diversification of electric-powered vehicles. Lithium is also prescribed as a medication against bipolar disorder. While Li can exert a toxic effect on living organisms, few studies have investigated the impact of anthropogenic inputs on Li levels in the environment. Here we report Li concentrations and Li isotope compositions of river, waste and tap water, and industrial products from the metropolitan city of Seoul. Results show that the large increase in population density in Seoul is accompanied by a large enrichment in aqueous Li. Lithium isotopes evidence a major release from Li-rich materials. Water treatment protocols are also shown to be inefficient for Li. Our study therefore highlights the need for a global Li survey and adequate solutions for minimizing their impact on ecosystems and city dwellers.

[1] Division of Earth and Environmental Sciences, Korea Basic Science Institute, Chungbuk 28119, South Korea. [2] Department of Science Education, Ewha Womans University, Seoul 03760, South Korea. [3] Department of Earth and Environmental Sciences, Pukyong National University, Busan 48513, South Korea. [4] Oceanography Laboratory of Villefranche-sur-Mer (LOV), CNRS, Sorbonne University, 06230 Villefranche-sur-Mer, France. *email: jongsikryu@gmail.com

During the last two decades, industrial demands for lithium (Li) resulted in a dramatic increase in Li production and, in 2017, the world production of Li from minerals and brine was 43,000 t[1]. As the secondary Li-ion battery (LIB) is a major and growing industrial channel for the element, approximately 660 million cylindrical Li-ion cells were produced in 2012, of which Korea shared 21% of total LIB manufacturing capacity[2]. Lithium is also incorporated into alloys and is widely used as a therapeutic drug for treating bipolar disorder since its discovery in 1970[3]. Although future demands will continue to grow and Li recycling may become an integral part of Li business[4], there are still few disposal process guidelines for waste LIB. Furthermore, there is a gap in our knowledge concerning the impact of these materials on Li levels in the environment as well as in municipal waters. The biological effect of high Li levels on the diet of several organisms and human beings has however been already reported in several publications[5–10]. For aquatic organisms, most of published studies have shown that elevated aqueous Li levels induce toxic effects[11,12]. Concerning humans, a growing number of studies have reported an inverse relationship between Li concentrations in drinking water and suicide mortality indices (in the USA, Japan and Lithuania), consistent with its biological role in brain cells[13–15]. In contrast, elevated Li concentrations in drinking water may be deleterious and disturb Ca homeostasis during pregnancy[16]. Interestingly, Li isotopes (the ratio of $^7Li/^6Li$) have been used by Earth scientists and geochemists since, when measured in rivers and soils, they provide key information on soil sustainability and weathering rate on continents, and therefore on the carbon cycle. They are considered as a key isotope proxy of unraveling why and how global climate could be regulated over geological timescale[17–22]. Thus, for all these reasons, it becomes increasingly important and urgent to quantify the amount of environmental Li that comes from anthropogenic activities. However, determining the conditions under which Li concentration or Li isotope signature can be impacted by anthropogenic activities remains a challenge.

Here, to test the effects of anthropogenic activities, we sampled and analyzed different types of water from the Han River (HR) basin. This river is the largest river system in South Korea, in terms of discharge and drainage area, and drains the Seoul Special Metropolitan City (Seoul), the capital and largest metropolis of South Korea. The population of the HR basin is estimated to be 12 million (Supplementary Table 1), of which more than 82.7% live in Seoul. Thus, this basin offers a unique opportunity to compare the upstream-inhabited part (although characterized by several dams) with the area of Seoul, located downstream of the Paldang Dam, and which is strongly impacted by urban and industrial activities. The downstream section of the Han River is also the main source of tap water for Seoul citizens. Our study provides the first Li isotope data of industrial products, allowing us to explain the significant Li-enrichment measured in the wastewaters, as well as the high Li contents in the Han River and tap water collected in the highly populated agglomeration of Seoul.

## Results and Discussion
**Water lithium and its isotopes upstream of the city of Seoul.** Compared to other rivers worldwide, the upper HR (HR1 in Fig. 1) and its two major tributaries (the Bukhan River, BR; and the Namhan River, NR) carry small amounts of dissolved Li ranging from 15.9 nM to 114 nM (see Methods section; Supplementary Table 2). This amount of Li is 2 to 16 times lower relative to the estimated global flow-weighted average (265 nM)[17]. Dissolved Li concentrations in each tributary are slightly variable over ~300 km down to the limit of Seoul city (Fig. 2). Lithium concentrations in the NR are systematically higher than those measured in the BR (Fig. 1), perhaps due to the occurrence of Li-rich shales. The lithium isotope compositions of these tributaries are significantly enriched in the heavy isotope ($^7Li$), with $\delta^7Li$ values all greater than 25‰. This finding is consistent with known mechanisms that fractionate Li isotopes during silicate weathering, such as $^6Li$-rich clay formation in soils[19–23]. The lithium isotope composition of both the BR and NR remains remarkably constant over ~340 km. This finding confirms the negligible impact of lithology on riverine $\delta^7Li$ values, as generally found in rivers of mixed lithology basins, since the major source of riverine lithium remains weathering, and leaching of silicate rocks and minerals[24,25]. Although the topography and runoff are slightly different in both watersheds (Fig. 1), these differences do not result in significant differences in Li isotope compositions, suggesting that, on average, the leaching and neoformation rate in soils are roughly equivalent over the whole watershed[26]. After the confluence of both tributaries, Li concentrations and Li isotope compositions of the HR upstream and downstream of the Paldang Dam show the negligible impact of this dam on Li, through its water regulation system.

**Evidence for strong anthropogenic Li input downstream.** In contrast to the upper watersheds, where all water Li levels are low ($50.4 \pm 29.2$ nM, $1\sigma$, $n = 14$), and the $\delta^7Li$ values are high and constant ($31.4 \pm 3.9$‰, $1\sigma$, $n = 14$), the downstream part of the HR basin displays a strong and progressive evolution for both parameters (Fig. 2; Supplementary Table 2). When the HR crosses Seoul from East to West, Li concentrations abruptly increase by a factor of 6, while $\delta^7Li$ values decrease significantly from 30.1‰ to 19.2‰. Both the changes in Li concentration and in $\delta^7Li$ covary with large increase in the population density, which passes from 5 million people at the HR2 site (just after the Paldang Dam) to more than 14 million people at the HR4 site (Supplementary Table 1). This relationship suggests that anthropogenic activities related to increasing urban activities are responsible for the changes displayed by the HR.

The influents correspond to wastewaters coming from households, hospitals and industries within the city, and ultimately arriving at the wastewater treatment plants (WWTP). The effluents correspond to waters treated with various methods to minimize their impact on the environment, and drained back to the river (Supplementary Fig. 1). The Han River also represents the major reservoir of drinking (tap) waters, which are used by consumer households after rigorous purification processes (Supplementary Fig. 2). Thus, any component enriched in wastewaters can affect both the Han River and tap waters. Interestingly, there is no significant difference between influent and effluent wastewaters for both Li concentrations and Li isotope compositions (Fig. 3). This finding demonstrates the negligible effect of the various water treatment protocols used in these plants on the Li level and its isotope composition in waters[27]. At present, the classic treatment systems are not adapted to Li pollution since there is no significant removal of this element during water treatment.

If the first striking result is that all effluent wastewaters (leaving the treatment plants) are strongly enriched in Li (up to >1 mM), the second is that their $\delta^7Li$ values are low ($14.5 \pm 4.3$‰, $1\sigma$, $n = 6$) and may therefore explain the decrease of the $\delta^7Li$ value displayed by the HR in Seoul (Supplementary Table 3). As shown in Fig. 4, the relationship between Li concentration and Li isotopes can be explained by the release of isotopically light Li from WWTP. This appears consistent since the only landfill of the area (the Sudokwon landfill, Fig. 1b) is located at 35 km West of Seoul and its drainage waters cannot contribute significantly to

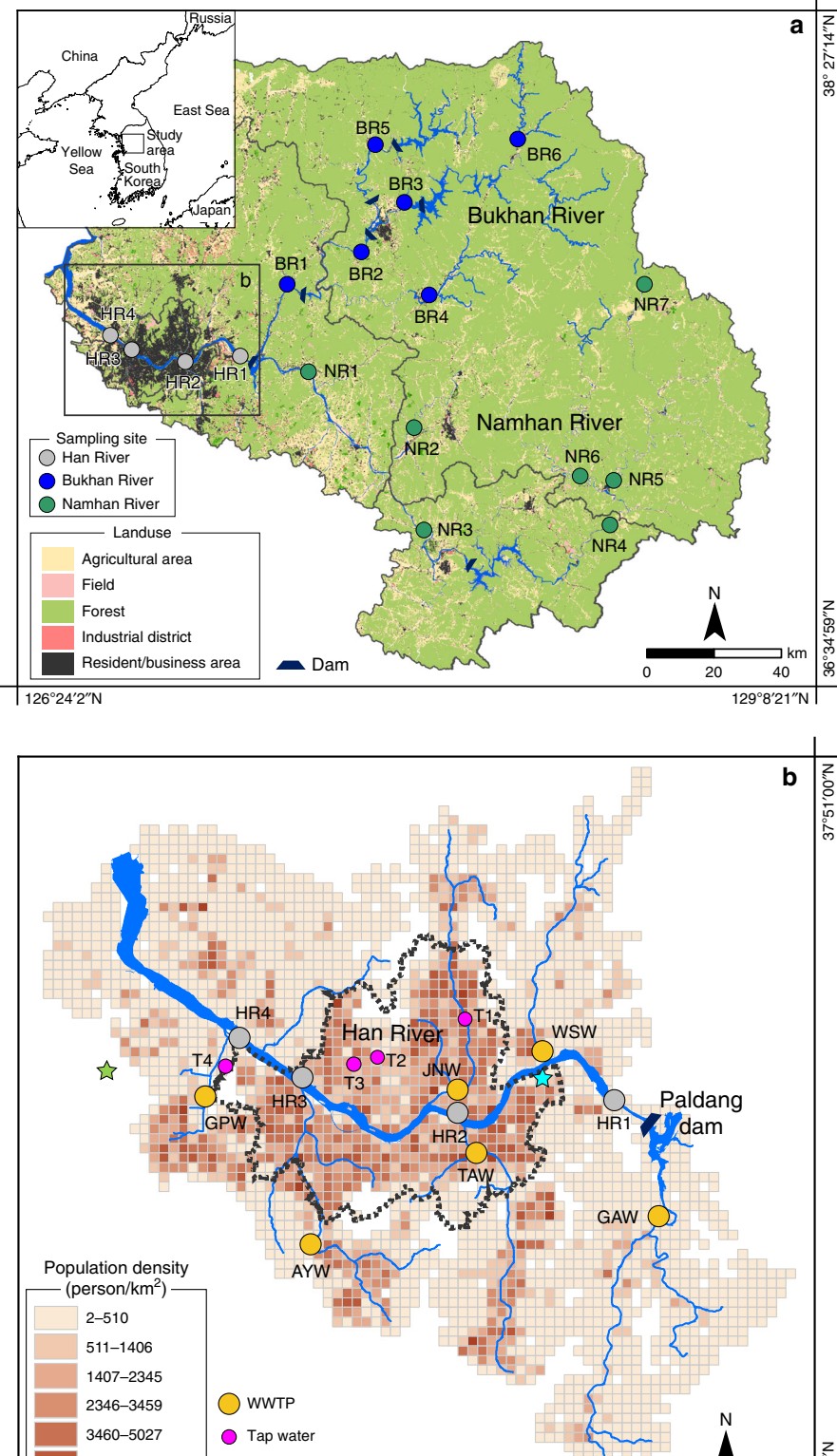

**Fig. 1** Map of the study area. Study area showing land use **a,** and population density and sampling sites **b**. Gyeongan (GAW), Wangsuk (WSW), Tan (TAW), Jungnang (JNW), Anyang (AYW), and Gulpo (GPW) display the location and name of wastewater treatment plants at which the wastewater was collected. Note that the HR4 site is located at ~30 km distance from the coastline.

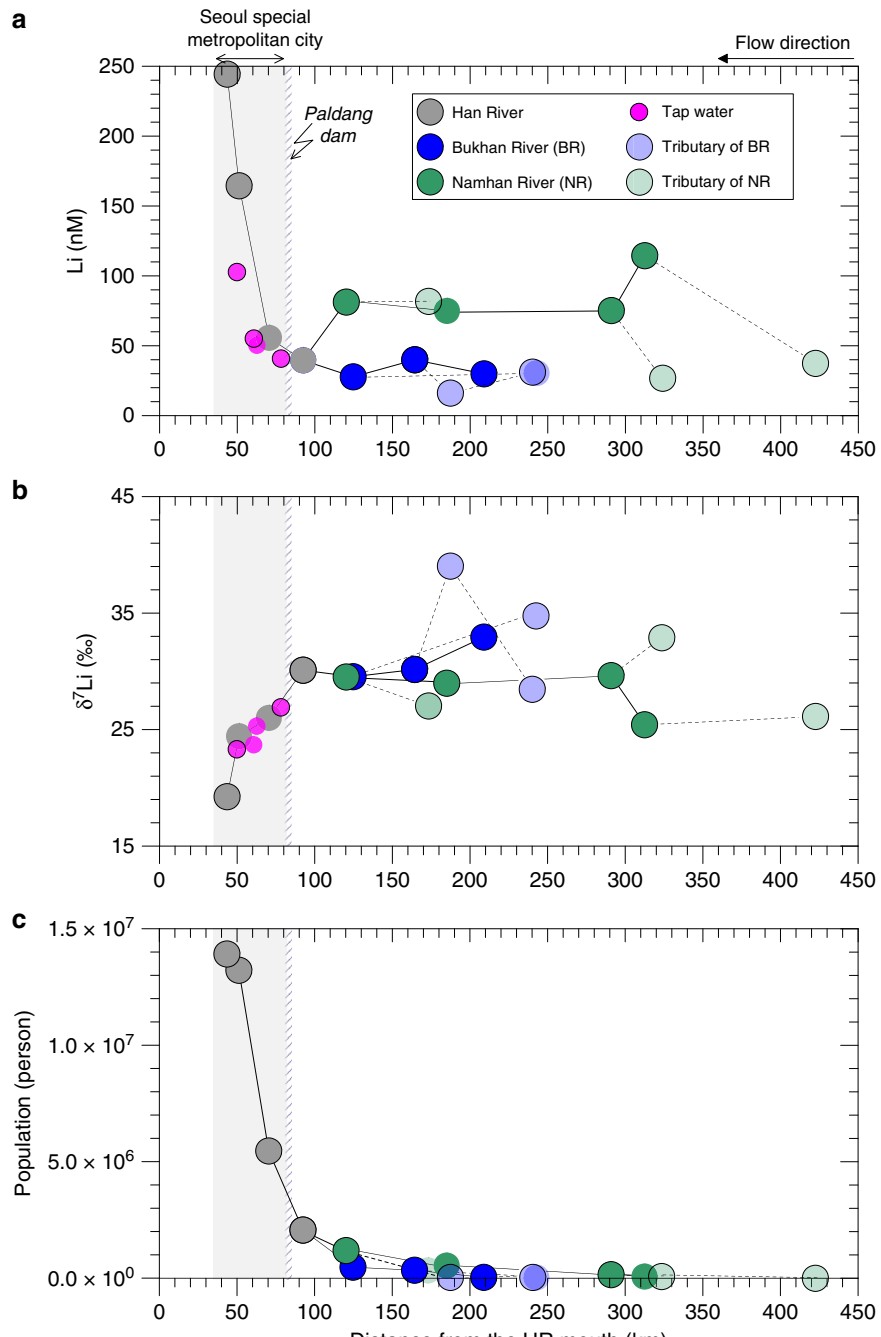

**Fig. 2** Relationship between dissolved lithium and population. Spatial variation in Li concentration **a** and Li isotope composition **b** measured in river water and tap water as a function of the distance from the Han River mouth. Variation of population living in the HR basin **c** as a function of the distance from the Han River mouth. Half-transparent circles represent the tributary of each river.

our samples. Lithium concentrations of both the downstream HR waters and the effluent wastewaters correlate positively with the population density, especially when it approaches 100 person km$^{-2}$ (Supplementary Fig. 3), supporting the link between population and volume of treated wastewater per unit population. We observe that this influence is more visible when the population density exceeds a certain threshold, typically higher than 100 person km$^{-2}$, and that there is also an influence of the effluent water discharge rate from each WWTP on the HR Li level (see Supplementary Fig. 4).

We analyzed several tap water samples collected in Seoul Special Metropolitan City (Supplementary Table 4; Fig. 1). As

shown in Fig. 2, Li concentrations and $\delta^7$Li values of tap water follow the same evolution from East to West, and are consistent with the values measured in the HR sampled in the same area. This finding strongly suggests that tap water is influenced by the same anthropogenic sources as the river, and confirms that the purification and wastewater treatment processes neither significantly lower Li level nor bias Li isotope composition. Altogether Li concentration and Li isotope composition show a binary mixing between a natural end-member (characterized by tributaries – BR and NR – draining rocks and soils upstream) and an anthropogenic end-member, consistent with isotopically light and strongly enriched wastewater (Fig. 4).

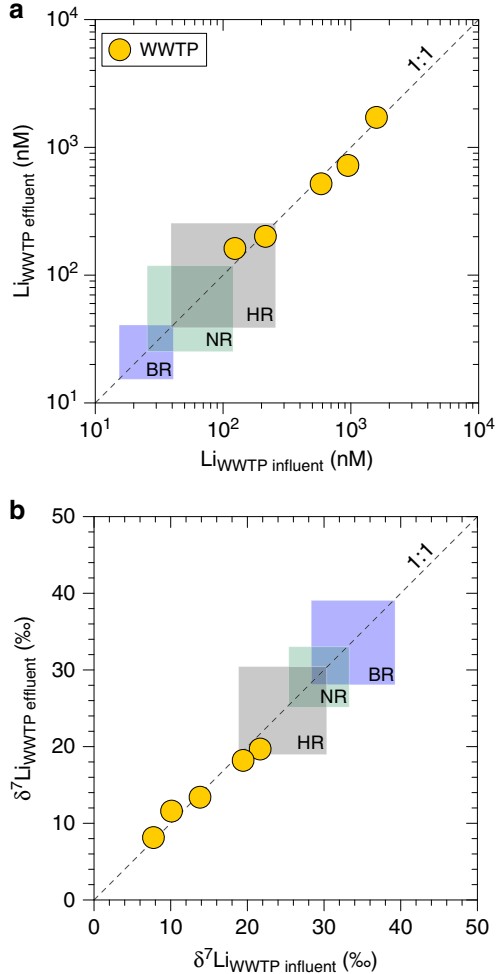

**Fig. 3** Lithium relationships in wastewaters. Li concentrations **a** and Li isotope compositions **b** of wastewater flowing in (influent) and flowing out (effluent) of wastewater treatment plants (WWTP, yellow circles). For comparison, the range of values obtained for the Han River and tributaries are also given (in squares).

Although the total population of mobile phone subscribers in South Korea was >43 million in 2015 (i.e., 84% of total population in South Korea)[28], only 1% of mobile phones were either exported or treated, due to no extended producer responsibility (EPR) regulations for LIB in Korea[29–31]. Therefore, it is likely that the high Li levels measured in waters would come from the release from LIB waste, along with other anthropogenic inputs such as therapeutic drug (Li carbonate), detergent and compost. In order to investigate further this possibility, we collected and analyzed several anthropogenic materials (Supplementary Table 5; Supplementary Figs. 3 and 4). As expected, the most enriched materials are the therapeutic drug, which contains about 10 wt% Li, and the LIB, which contain between 4.1 and 7.6 wt% Li. Both display systematically low $\delta^7Li$ values, ranging from 2.4‰ to 13.3‰, consistent with the low $\delta^7Li$ values displayed by the wastewaters. The other analyzed materials contain much less Li (0.53–2.92 µg g$^{-1}$ d.w. for the detergents and <0.39 µg g$^{-1}$ d.w. for the food wastewater and compost) and display, on average, slightly heavier $\delta^7Li$ values (15.6‰). Thus, as shown in Fig. 4, this first isotopic investigation of Li-rich materials allows us to explain both the significant Li-enrichment of wastewaters and their low $\delta^7Li$ values. Since treated and untreated waters are similarly enriched in Li, and explain the decrease of river $\delta^7Li$ in Seoul when the population density is high (Figs. 2c and 4), Li isotopes

confirm a major impact of the use of anthropogenic products on Li levels in river crossing the city and in municipal waters. Overall, our study shows that the large Li inputs observed in the Han River come from LIB, therapeutic drug, and food waste, all likely proportional to the population, combined with the inefficiency of wastewater treatment for Li-removal.

Compared to other trace metals[32,33], such as Zn, Cu, Ni or Hg, for which contaminated zones are clearly identified and monitored, and whose impacts on aquatic organisms and plants have been carefully investigated for many years in ecotoxicology, there is little information on environmental Li and its toxic effect. By illustrating anthropogenic Li inputs in Seoul waters, our study highlights the need to estimate the environmental and health impact of Li-rich materials, particularly in highly populated areas. Understanding the biological and metabolic effects of high Li levels on aquatic ecosystems also remains to be investigated to fill the gap compared to other contaminants. Finally, this study highlights that in urban areas, Li isotopes are more sensitive to anthropogenic inputs rather than local weathering inputs and therefore should be used with caution as a weathering proxy.

## Methods
**Samples collection and field measurements.** We collected 27 samples in July 2015 from 22 sites along a 422 km downstream transect between the uppermost reaches and the estuary of the Han River (Fig. 1). Upstream, the HR includes two major tributaries (i.e., the Bukhan River, BR; and the Namhan River, NR), which join at the Paldang Dam to form the main channel of the HR that crosses the capital city downstream. Due to the large population in Seoul, there are six major WWTPs that flow out to the HR. The wastewater going in (influent) and going out (effluent) of these plants was collected to estimate the impact of water treatment protocols on dissolved Li. Finally, we compared our results to several tap water samples from various locations in Seoul (Fig. 1) and to several anthropogenic sources. Sample locations were documented with a Garmin GPSMAP 60CSx handheld GPS meter. Temperature (±0.1 °C) and pH (±0.002) were measured in situ using an Orion 5-STAR portable meter equipped with an Orion 3-in-1 pH/ATC pH electrode. The electrode was calibrated twice per day using pH = 4.01, 7.00, and 10.01 buffers. Samples for dissolved cations, trace elements, and Li isotope measurements were passed through 0.2 µm filters, collected in I-CHEM LDPE bottles, and acidified to pH = 2 using concentrated, ultrapure HNO$_3$. Samples for dissolved anions and total alkalinity ($A_T$) were passed through 0.2 µm filters and collected in I-CHEM LDPE bottles.

**Chemical analysis.** Cation and trace element concentrations were measured using a Perkin Elmer Optima 8300 ICP-AES and a Thermo Elemental iCAP$^{TM}$ Q ICP-MS at the Korea Basic Science Institute (KBSI). Analyses of NRCC SLRS-4 and CRM TMDW-A were within ± 5% of certified values. Anion concentrations were measured using a Dionex ICS-1100 ion chromatograph equipped with a Dionex$^{TM}$ IonPac$^{TM}$ AS14 anion-exchange column. The total carbonate alkalinity in µeq/L ($A_T = HCO_3 + 2CO_3$) was measured using a Mettler Toledo T50A titrator with 0.01 M HCl acidimetric titration to an endpoint of pH = 4.5. The percent charge balance error (CBE), as one measure of the data quality, is given by the equation [CBE (%) = (TZ$^+$ − TZ$^-$)/(TZ$^+$ + TZ$^-$) × 100], where TZ$^+$ = 2Ca$^{2+}$ + 2Mg$^{2+}$ + K$^+$ + Na$^+$, TZ$^-$ = Cl$^-$ + 2SO$_4^{2-}$ + NO$_3^-$ + A$_T$, and is on average better than ± 2% (Supplementary Table 2).

**Lithium isotope analysis.** Samples containing ~100 ng Li were dried in Teflon vessels, and the residues were treated with concentrated HNO$_3$, dried, and re-dissolved in a 1:4 (v/v) mixture of 6 M HNO$_3$ and 100% methanol. Lithium was separated from matrix elements using an AG 50 W–X8 resin (200–400 mesh)[26]. Then, the sample was dried and re-dissolved in 5% HNO$_3$ (~40 ppb Li). Lithium isotope ratios were measured using a Neptune MC-ICP-MS upgraded with a large dry interface pump at the KBSI and the Korea Institute of Ocean Science & Technology (KIOST). Samples were introduced using a quartz dual cyclonic spray chamber and analyzed with a blank-standard-blank-sample-blank-standard-blank bracketing method. Sample intensities were matched to within 10% of the intensity of the standard. The sensitivity was ~90 V ppm$^{-1}$ on mass 7 at a typical uptake rate of 100 µL min$^{-1}$. Prior to isotopic analysis, each sample was checked for the yield and the concentration of matrix elements. The yields were approximately 100%, and the matrix concentration did not exceed 1.5% of the Li concentrations. The lithium isotopic composition is reported in delta notation relative to NIST RM 8545, where $\delta^7Li = [(^7Li/^6Li)_{sample}/(^7Li/^6Li)_{NIST\ RM\ 8545} − 1] × 1000$. The accuracy and reproducibility of the whole method was validated using the USGS rock reference materials (BCR-2, BHVO-2, and BIR-1) and seawater standard (IAPSO). BCR-2 yielded +3.6 ± 1.7‰ (2σ, $n = 14$), BHVO-2 yielded +4.5 ± 0.0‰ (2σ,

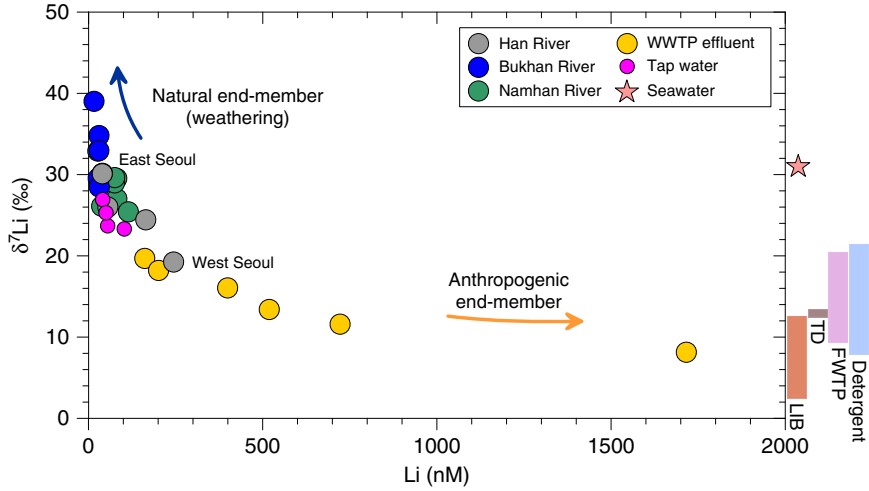

**Fig. 4** Li concentrations versus Li isotope compositions. Wastewater (yellow) plots towards a Li-rich anthropogenic end-member, while the Han River tributaries (BR in blue and NR in green), sampled upstream of the basin, plot towards a natural end-member, consistent with fractionating mechanisms during water–rock interactions (soil/rock weathering). The Han River crossing Seoul from East to West (in grey) evolves progressively towards the anthropogenic end-member represented by wastewaters and by the various Li-rich materials. LIB, TD and FWTP represent secondary Li-ion battery (LIB), therapeutic drug (TD) and food waste treatment plant (FWTP), respectively. Tap waters (in pink) follow the same trend and are consistent with the HR water from which they are sourced. Seawater data were taken from refs. [39,40].

$n = 2$), BIR-1 yielded +4.1‰ ($n = 1$), and IAPSO yielded +31.2 ± 1.5‰ (2σ, $n = 15$), which were all in good agreement with reported values[21,34–38].

**Data availability**

All data generated or analyzed during this study are included with this published article in its Supplementary Information.

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

## Acknowledgements

We thank Y.K. and K.R. for providing cathode materials for LIB and MC-ICP-MS at the KIOST. This project was supported by the National Research Council of Science & Technology (NST) grant by the Korea government (MSIP) (No. CAP-17-05-KIGAM) and also benefited from discussions initiated in the context of the ANR ISO2MET Grant (ANR-18-CES34-0002) started in January 2019.

## Author contributions

J.-S.R and N.V. designed the study, and led the writing of the manuscript. H.-B.C., W.-J.S. and J.-S.R. conducted the fieldwork and chemical analyses. All authors contributed equally to the data interpretation.

## Competing interests

The authors declare no competing interests.
