## [Peer Review File · Nature Communications]

Reviewers' comments:

Reviewer #1 (Remarks to the Author):

Choi and co-workers present a study on Li concentration and isotopic composition in the larger Seoul watershed. They document rather interesting trends: Li concentrations increase as the Han River crosses Seoul and Li isotopic composition becomes enriched in the lighter isotopes. The authors identify waste water treatment plants as likely sources, and show that Li isotopes in technological Li applications, i.e. batteries, correspond to the anthropogenic Li isotope end-member.

The study is rather elegant in its simplicity, and pertinent in terms of the environmental awareness that it raises. Results are not overstated, the methods are detailed and of high quality. Since I am not an expert on lithium, I have spent an hour or so on the web of science to see what studies exist on Li pollution. I confirm the author's statements in the introduction, which is 'not much'. This would be the first environmental application of Li pollution tracing using Li isotopes. I therefore think it would of interest to the broader environmental chemistry and biogeochemistry community. And I therefore recommend publication in Nature Communications.

Jeroen Sonke, Toulouse, May 2019

Minor comments:

L17 "Our study therefore unravels the need to perform a lithium survey..."

One can unravel some complex problem....but not a need. Suggest to write 'Our study highlights the need for a lithium survey at the global scale...'

L29 and 34. 'Few studies have been published....toxic', and 'Li isotopes are widely used by earth scientist...'

Few vs widely....each phrase has 5 citations. Suggest to delete 'widely'.

L39. « Determining the conditions under which its concentration or its isotope signature can be artificially biased becomes a challenge. »

What is meant by artificially biased? Anthropogenically impacted?

Not clear in Fig 2 caption what the smaller, half-transparent green and blue circles represent?

L91. "...Li isotope compositions are high and constant..."

It is not the isotope composition that is high....it is the the $d7Li$ parameter (or $7/6Li$ ratio) that is high.

If space permits I would suggest to move Figure 1 to the main text. It is very convincing in terms of binary mixing.

Reviewer #2 (Remarks to the Author):

The aim of this manuscript is to study and quantify the amount of Li in the Han River basin (Korea) that is sourced from industrial anthropogenic activities. The production and usage of Li has significantly increased over the past two decades owing to both the electronic revolution and energy transition. Despite recognition of this, few studies have investigated the source and fate of anthropogenic-derived lithium in urban environments. In this study, the authors have combined Li concentration and isotope measurements in river water, tap water and wastewater treatment plant samples with geographical and geological information. They found that the concentration of Li in the

Han River increases by a factor of 6 when crossing the city of Seoul, and this increase is accompanied with a decrease of the dissolved Li isotope composition. Because the Li-batteries have low Li isotope composition and the Li concentration and isotope composition of river samples is correlated with the population density, they suggest that “anthropogenic activities related to the number of inhabitants are responsible for the changes displayed by the Han River” (lines 98-99), through the disposal of Li-batteries.

I think that the purpose of this study is novel, timely and of interest to a wide range of people and scientific communities. The dataset coverage is good and the observed trends are quite spectacular. Clearly, as justified by the authors, there is a major input of anthropogenic Li into the Han River when it crosses the city of Seoul and this has major implications for environmental policies. However, I find that some aspects of the discussion about the origin, characterization and control of this anthropogenic source(s) of Li are problematic and not very clear. For this reason, I suggest that this manuscript could be considered for publication in Nature communication providing major revisions are carried out. Below, I detail my main criticisms of the manuscript.

The authors suggest that the high Li content is related to the high population density (title of the paper, Lines 98-99, correlation with population density Fig. 4) through the release of Li from Li-batteries due to the absence of disposal process guidelines for waste batteries (Line 111), but they do not provide or explicitly discuss a causal mechanism. How is the Li transferred from the battery into the river? (this is not discussed at all in the manuscript). Why would the population density control the Li release? Is it because of widespread disposal of used Li-batteries by inhabitants? (i.e. each inhabitant has electronics with Li-battery so the more inhabitants, the more disposal of used batteries, the more contamination of the water due to the leaking of disposed Li batteries?). The authors need to discuss or clarify these aspects.

Secondly, similarly to Li, the concentrations of most of the major anions and cations are significantly increasing downstream. What processes or input sources are responsible for this concentration increase? Since these elements are not present (or in very low amounts) in Li batteries, it shows that there are possibly other sources of dissolved ions to the river. Could these sources potentially account for some of the observed increase Li concentration? It is possible to partly answer this question by normalizing the Li concentrations (e.g. to Cl or Na). When Li concentrations are normalized, it appears that only the wastewater effluent “TAW” (and not the others effluents) is a significant source of Li relative to other elements and relative to the upstream non-impacted tributaries. Clearly, there is a specific local source of Li only in that part of the city (corresponding to TAW), and this input explains most of the observed Li concentration increase downstream. This lead to my third criticism of the discussion: I find the correlation between the Li concentration and the population density (Fig. 4a) misleading. The highest Li concentration (by far) and Li/Cl (or Li/Na) ratio do not correspond to the area with the highest population density (WWTP effluent “TAW”, Fig. 4). The anthropogenic input of Li to the river is local and not widespread, and does not correspond to the zone of highest population density. In my opinion, this contradicts the title of the manuscript, and gives some indications about the source of this anthropogenic Li. Is there something specific in this area relative to other areas of the city that could potentially explain the observed high Li content? I understand that it is beyond the scope of this study to precisely characterize all the potential source of anthropogenic Li in urban environments, and the authors mention several possible sources of Li. Nevertheless, I think that data interpretation could be expanded with the above suggestions. As a final general comment, Seoul is close to the estuary, could the tide have an influence on the river chemistry?

Below are more specific comments:

- Line 1: the title is problematic as the highest source of Li is not where the population density is the highest
- Lines 57-59: what is the “drainage” area covered by these wastewater treatment plants and how is the population density calculated for these samples (Fig. 4).
- Line 136: I think Figure S1 should be in the main text instead of Figure 3.

Reviewer #3 (Remarks to the Author):

The manuscript presents concentration and isotope ratio data for Li from surface, waste, and tap water from around a major city. The goal is to begin to understand the scope of Li contamination in municipal waters. Lithium is an element frequently not quantified in surface and drinking waters, so there is something of a gap in our knowledge of the baseline of Li in these waters.

I think the idea is an interesting one and one which merits investigation by the geochemical community. From that standpoint, I think the manuscript has a certain value to a Nature audience. Although the health impacts of dissolved Li are far from clear, most studies do not signal immense concern for Li toxicity in general. One of the pivotal points to this manuscript is that dissolved Li might become a toxicity concern, so the hydrochemical understanding is important to a global urban population. I am not convinced of that being a driving force enough to make the work of the highest impact.

I think the data support the inferences the authors make in a general way. However, this is a far from airtight case. Many questions lay behind the small data set presented here, some which might be dispatched with some more thorough explanations, but others need more data before they can be confronted.

The manuscript uses population density and WW effluent in developing interpretations, which suggests a link between population and number of WWT facilities or volume of treated WW per unit population. That link is not mentioned in the text. Furthermore, although some waste Li would be “flushed” waste (and hence be part of municipal WW treatment), Li from battery or other solid waste would likely come from significantly from leakage from landfills. Distribution of these is also not mentioned in the text. Landfills are mentioned (line 116), but it is implied that water from landfills would be part of the WW treatment stream—unless Korean municipal water treatment differs from N. America, this is not likely to be so. Some description of what the water treatment protocols are would be helpful. How does water treatment for drinking water differ from that of wastewater (in most places these streams are separate)?

Although I think the study is one which, with some more data and clarification, is very valuable to a variety of audiences, I think that in its current form it poses too much speculation.

line 50: I wonder how much secular variation occurs in this system. Basically is one month’s worth of data a reasonable sample from which we can expect lasting interpretations to come?

line 55: The data themselves are from ArcGIS? I should think the data were brought in from individual sources and the manipulations were done with ArcGIS (see also Table S1).

line 74: This is unclear: shales are typically among the most Li-rich rocks.

line 78: replace “formations” with “formation”

line 105: replace “starts to” with “approaches”

line 140, 147: replace “anthropic” with “anthropogenic”

Fig. 2: What is the difference between large and small symbols?

**Reply letter to the Reviewers' comments:**

Reviewer #1 (Remarks to the Author):

Choi and co-workers present a study on Li concentration and isotopic composition in the larger
Seoul watershed. They document rather interesting trends: Li concentrations increase as the Han
River crosses Seoul and Li isotopic composition becomes enriched in the lighter isotopes. The
authors identify waste water treatment plants as likely sources, and show that Li isotopes in
technological Li applications, i.e. batteries, correspond to the anthropogenic Li isotope end-
member.

The study is rather elegant in its simplicity, and pertinent in terms of the environmental
awareness that it raises. Results are not overstated, the methods are detailed and of high quality.
Since I am not an expert on lithium, I have spent an hour or so on the web of science to see what
studies exist on Li pollution. I confirm the author's statements in the introduction, which is 'not
much'. This would be the first environmental application of Li pollution tracing using Li
isotopes. I therefore think it would of interest to the broader environmental chemistry and
biogeochemistry community. And I therefore recommend publication in Nature Communications.

Jeroen Sonke, Toulouse, May 2019

We thank this reviewer for his positive review about our manuscript as it is indeed the first
investigation demonstrating that Li levels and Li isotopes in river waters can be significantly
impacted by anthropogenic inputs. Please see our response to his minor comments just below.

**Minor comments:**

L17 "Our study therefore unravels the need to perform a lithium survey..."
One can unravel some complex problem....but not a need. Suggest to write 'Our study highlights
the need for a lithium survey at the global scale...'

**Response:** We agree and revised the text as commented.

**Action:** On lines 9 – 11, we revised the text to read: "Our study therefore highlights the need for
a global Li survey...".

L29 and 34. 'Few studies have been published....toxic', and 'Li isotopes are widely used by
earth scientist...'

Few vs widely....each phrase has 5 citations. Suggest to delete 'widely'.

**Response/Action:** We have added some references to the text but please note that we are limited
in reference number. Also, on line 29, we deleted 'widely'.

L39. « Determining the conditions under which its concentration or its isotope signature can be

artificially biased becomes a challenge. »

What is meant by artificially biased? Anthropogenically impacted?

**Response/Action:** On lines 34 – 36, we corrected this in the text as follows: “However,
determining the conditions under which Li concentration or its isotope signature can be impacted
by anthropogenic activities remains a challenge.”

Not clear in Fig 2 caption what the smaller, half-transparent green and blue circles represent?

**Response/Action:** We added what they represent in the revised Figure caption and in its legend.

L91. “...Li isotope compositions are high and constant...”

It is not the isotope composition that is high....it is the the $d7Li$ parameter (or $7/6Li$ ratio) that is
high.

**Response/Action:** On line 84, we replaced ‘the Li isotope compositions’ with ‘the δ^7Li values’

If space permits I would suggest to move Figure 1 to the main text. It is very convincing in terms
of binary mixing.

**Response/Action:** We agree with the reviewer suggestion and have added this figure to the main
text (Figure 4).

Reviewer #2 (Remarks to the Author):

The aim of this manuscript is to study and quantify the amount of Li in the Han River basin
(Korea) that is sourced from industrial anthropogenic activities. The production and usage of Li
has significantly increased over the past two decades owing to both the electronic revolution and
energy transition. Despite recognition of this, few studies have investigated the source and fate of
anthropogenic-derived lithium in urban environments. In this study, the authors have combined
Li concentration and isotope measurements in river water, tap water and wastewater treatment
plant samples with geographical and geological information. They found that the concentration
of Li in the Han River increases by a factor of 6 when crossing the city of Seoul, and this
increase is accompanied with a decrease of the dissolved Li isotope composition. Because the Li-
batteries have low Li isotope composition and the Li concentration and isotope composition of
river samples is correlated with the population density, they suggest that “anthropogenic
activities related to the number of inhabitants are responsible for the changes displayed by the
Han River” (lines 98-99), through the disposal of Li-batteries.

I think that the purpose of this study is novel, timely and of interest to a wide range of people and
scientific communities. The dataset coverage is good and the observed trends are quite
spectacular. Clearly, as justified by the authors, there is a major input of anthropogenic Li into
the Han River when it crosses the city of Seoul and this has major implications for environmental
policies. However, I find that some aspects of the discussion about the origin, characterization
and control of this anthropogenic source(s) of Li are problematic and not very clear. For this
reason, I suggest that this manuscript could be considered for publication in Nature
communication providing major revisions are carried out. Below, I detail my main criticisms of

the manuscript.

We thank the reviewer for his/her positive comments.

The authors suggest that the high Li content is related to the high population density (title of the
paper, Lines 98-99, correlation with population density Fig. 4) through the release of Li from Li-
batteries due to the absence of disposal process guidelines for waste batteries (Line 111), but they
do not provide or explicitly discuss a causal mechanism. How is the Li transferred from the
battery into the river? (this is not discussed at all in the manuscript). Why would the population
density control the Li release? Is it because of widespread disposal of used Li-batteries by
inhabitants? (i.e. each inhabitant has electronics with Li-battery so the more inhabitants, the more
disposal of used batteries, the more contamination of the water due to the leaking of disposed Li
batteries?). The authors need to discuss or clarify these aspects.

**Response:** It is true that we were not presenting clearly in the manuscript the direct link that
exists between battery or other Li-rich materials, wastewaters, river and tap waters. This is now
done, as explained below in three parts:

109 A. We have added two key schematic diagrams explaining clearly how wastewaters are
110 related to the Han River, and how the Han River is related to tap waters (after corresponding
treatments) in the Supplementary Information. Note that there is only one landfill site (the
Sudokwon Landfill) in the Seoul locality, at 35 km west of Seoul (now shown in Fig. 1).
Therefore, the leachate from this landfill does not affect significantly our river samples.

**Action:** This is now made clearer in the text, in the revised Fig. 1 and in the Supplementary
Information.

B. The Korea Ministry of Environment promulgated the extended producer
responsibility (EPR) regulation for a number of electrical and electronic products in 2003 so that
66% of primary lithium batteries were recycled in 2015 (<http://www.kbra.net/epr/epr5.htm>).
However, secondary Li-ion battery (LIB) has not been included in the EPR list so that the
volume of LIB collected and recycled through the EPR system is small. LIBs are commonly used
in mobile phones (44%), laptop computers (27%) and table PCs (6%), and the waste of those
electronics can act as anthropogenic source of lithium. In 2015, total population of mobile phone
subscribers was more than 43 million, i.e. 84% of total population in South Korea (the Korea
Ministry of Science and ICT; <http://www.itstat.go.kr>). The sales volume of mobile phones is 15
million units every year and the estimated volume of obsolete mobile phones accounts for about
80% of the sale volume (Lee et al., 2007). Only about 4 million units of waste mobile phones
(=1%) were collected by Korean telecommunication industries and were either exported (2
million units) or treated and recycled (Song, 2004; Lee, 2006). Therefore, these numbers
demonstrate the large number of LIB not being recycled and that the more inhabitants, the more
disposal of used LIBs.

**Action:** We added, some of the quantification concerning the use of LIB in the text on lines 129
135 – 132, and the new data in Tables S3 & S5.

C. In the first version of the manuscript, we explored only the Li Battery because of their
very high Li content. However, as we were also highlighting, Li may also come from other types
of Li-rich materials: therapeutic drugs, compost, and detergent. Since then, we have analyzed Li
concentrations and its isotope compositions for these products too. Therapeutic drugs used for
bipolar disorder (Li carbonate) display high Li content as expected (10.5 ± 0.6 wt.%, 1σ , $n=2$),
but also low $\delta^7\text{Li}$ value ($12.8 \pm 0.8\text{‰}$, 1σ , $n=2$). Also, the compost and two wastewater samples
coming from food waste treatment plant contain high Li levels ($55.8 \mu\text{M}$ and $2.90 \pm 0.45 \mu\text{M}$,
respectively) and more variable $\delta^7\text{Li}$ values (9.3‰ and $19.4 \pm 1.5\text{‰}$, respectively). Several
detergents commonly used in South Korea display high Li contents ($224 \pm 144 \mu\text{M}$, 1σ , $n=4$) and
also slightly higher $\delta^7\text{Li}$ values on average ($15.4 \pm 6.0\text{‰}$, 1σ , $n=4$).

Overall, the $\delta^7\text{Li}$ values of the most enriched materials (typically > 5 ppm Li dry weight; see
revised Table S5) that may contribute to the WWTP composition are low ($\delta^7\text{Li} < 13\text{‰}$). These
new investigations strongly support that the high Li levels of wastewaters can be explained by
the release from these materials, and that the negative correlation observed in Fig. 4 is best
explained by a strong input from these anthropogenic materials. This also very consistent with
the positive trend with population density (Figs. 1 & S3).

**Action:** We revised the manuscript accordingly, and added these new data to the text, Table S5,
and Figs. 4 & S3.

Secondly, similarly to Li, the concentrations of most of the major anions and cations are
significantly increasing downstream. What processes or input sources are responsible for this
concentration increase? Since these elements are not present (or in very low amounts) in Li
batteries, it shows that there are possibly other sources of dissolved ions to the river.
Could these sources potentially account for some of the observed increase Li concentration? It is
possible to partly answer this question by normalizing the Li concentrations (e.g. to Cl or Na).
When Li concentrations are normalized, it appears that only the wastewater effluent “TAW” (and
not the others effluents) is a significant source of Li relative to other elements and relative to the
upstream non-impacted tributaries. Clearly, there is a specific local source of Li only in that part
of the city (corresponding to TAW), and this input explains most of the observed Li
concentration increase downstream. This lead to my third criticism of the discussion: I find the
correlation between the Li concentration and the population density (Fig. 4a) misleading. The
highest Li concentration (by far) and Li/Cl (or Li/Na) ratio do not correspond to the area with the
highest population density (WWTP effluent “TAW”, Fig. 4). The anthropogenic input of Li to the
river is local and not widespread, and does not correspond to the zone of highest population
density. In my opinion, this contradicts the title of the manuscript, and gives some indications
about the source of this anthropogenic Li. Is there something specific in this area relative to other
areas of the city that could potentially explain the observed high Li content? I understand that it
is beyond the scope of this study to precisely characterize all the potential source of
anthropogenic Li in urban environments, and the authors mention several possible sources of Li.
Nevertheless, I think that data interpretation could be expanded with the above suggestions.

**Response:** We now report new data for other potential sources of Li to the wastewaters. Results
are shown in the revised Table S5 and in Figs. 4 & S3, and highlight two key points. First, some
of these materials can be enriched in major elements, such as Na or Ca (for detergents and

compost), but given the large amounts of these elements already present in the water, this may
not represent a significant impact, although it would deserve to be quantified more precisely. It is
a difficult to quantify precisely the contribution of the anthropogenic pollution to the Han River
by using Li/Cl or Li/Na since the Wastewaters display large variations, due principally to the
regulating use of Cl and Na for water chlorination (see new Fig. S2). The second point is that,
materials with the highest Li concentrations are isotopically light (typically $\delta^7\text{Li} < 13\text{‰}$; revised
Table S5), and this is consistent with the Wastewaters endmember shown in Figs. 4 & S3 since
all of them are significantly enriched in Li (650 nM on average) compared to the Han River
downstream (< 82 nM), and display lower $\delta^7\text{Li}$ values as well.

As the reviewer pointed out, the highest Li concentration and Li/Na ratio do not directly
correspond to the WWTP with the highest population density (noted “JNW” in Fig. S3). This
distinct feature of the “TAW” WWTP compared to the others may be due to the food waste
treatment plant that exists only in its drainage area (Fig. 1B), as we have shown that its released
wastewater contains high Li content (2.90 ± 0.45 μM , 1σ , $n=2$; revised Table S5). The discharge
rate of TAW is lower than that of JNW having the highest population density, which explains the
discrepancy with the population density observed for this sample (Table S3; Fig. S4).

**Action:** We added these new data and revised the main text and Supplementary Information by
adding WWTP discharge rate and Li/Na ratios in Fig. S4.

As a final general comment, Seoul is close to the estuary, could the tide have an influence on the
river chemistry?

**Response:** We thank the reviewer for this comment and we would like to point out that we had
initially explored this possibility. However, we think that we have two solid arguments against
this effect: first, seawater does not explain the Li enrichment and $\delta^7\text{Li}$ decrease observed in the
HR water because its $\delta^7\text{Li}$ value is high ($31.2 \pm 0.2\text{‰}$; Millot et al., 2004; see revised Fig. 4).
Second, it has been previously reported that the seawater intrusion does not occur in coastal
aquifers for the western coastal area of South Korea, which is about 4 km from the coastline (e.g.,
Park et al., 2005). Our river sampling site the closest to the estuary (HR4) is located at ~ 30 km
distance from the coastline.

**Action:** We added this in Fig. 1 and plotted the seawater value in the revised Fig. 4.

Below are more specific comments:

- Line 1: the title is problematic as the highest source of Li is not where the population density is
the highest

**Response:** We understand the reviewer’s comment but as we address in previous responses
above, high Li content and low $\delta^7\text{Li}$ value are closely related to population density (Fig. 2).

- Lines 57-59: what is the “drainage” area covered by these wastewater treatment plants and how
is the population density calculated for these samples (Fig. 4).

**Response:** Based on the information on the drainage area covered by each WWTP and
population from the Ministry of Environment (<http://www.me.go.kr>), we calculated the
population density as dividing population by the drainage area.

**Action:** We added the drainage area, population and population density in the revised Table S3.

- Line 136: I think Figure S1 should be in the main text instead of Figure 3.

**Action:** As it was also advised by the Reviewer #1, we moved Figure S1 to the main manuscript
(Fig. 4).

**References:**

- 1. Lee, J.-c., Song, H.T., Yoo, J.-M. Present status of the recycling of waste electrical and
electronic equipment in Korea. *Resour. Conserv. Recy.* **50**, 380-397 (2007).
- 2. Song, H.T. Current status of the recycling of waste mobile phones and urgent problem. *e-*
*Recycling* **8**, 7-9 (2004) (in Korean).
- 3. Lee, J.-c. Private communication with SK networks, <http://www.sknetworks.co.kr>. (2006).
- 4. Millot, R., Guerrot, C., Vigier, N. Accurate and high-precision measurement of lithium
isotopes in two reference materials by MC-ICP-MS. *Geostand. Geoanal. Res.* **28**, 153-159
(2004).
- 5. Park, S.-C. *et. al.* Regional hydrochemical study on salinization of coastal aquifers, western
coastal area of South Korea. *J. Hydrol.* **313**, 182-194 (2005).

Reviewer #3 (Remarks to the Author):

The manuscript presents concentration and isotope ratio data for Li from surface, waste, and tap
water from around a major city. The goal is to begin to understand the scope of Li contamination
in municipal waters. Lithium is an element frequently not quantified in surface and drinking
waters, so there is something of a gap in our knowledge of the baseline of Li in these waters.
I think the idea is an interesting one and one which merits investigation by the geochemical
community. From that standpoint, I think the manuscript has a certain value to a Nature audience.

Although the health impacts of dissolved Li are far from clear, most studies do not signal
immense concern for Li toxicity in general. One of the pivotal points to this manuscript is that
dissolved Li might become a toxicity concern, so the hydrochemical understanding is important
to a global urban population. I am not convinced of that being a driving force enough to make
the work of the highest impact.

**Response:** As underlined by the reviewer, there is a gap in our knowledge concerning the
baseline of Li in the environment, in particular in municipal waters, small and large rivers
draining cultivated regions, and in plants and animals. Thus far, in the literature there is not an
immense concern (although not negligible) for Li toxicity, and we think that our study
demonstrates that this must change in the future.

As summarized in our manuscript, high Li levels can be related either to positive aspects (less
suicides and for treating bipolarity), or to toxic and deleterious consequence (during pregnancy).

Several studies have advised Li to be considered as an essential element due to its biological role
but others have highlighted its high toxicity for aquatic species. In nature, Earth scientists have
neglected these aspects because of few measurements in plants that show low Li levels. However,
the number of data in organic samples is still scarce (except in marine fossils).

In geochemistry, Li isotopes are perhaps the most promising proxy unraveling why and how
global climate has been regulated by continental weathering over geological timescales. Recently,
large scale studies of continental weathering are based on river sampling performed in highly
populated areas (in Asia in particular).

For all these reasons, and given the increasing importance of this element in modern life, we
think that our study is of importance and of interest for different research fields (ecotoxicology,
geochemistry, climate, weathering, health & environment and water quality).

**Action:** On lines 20 – 32, we revised the text to better highlight these points.

I think the data support the inferences the authors make in a general way. However, this is a far
from airtight case. Many questions lay behind the small data set presented here, some which
might be dispatched with some more thorough explanations, but others need more data before
they can be confronted.

**Response/Action:** Please see our detailed response to Reviewer #2. In brief, we conducted
additional field works in order to collect samples representing different potential Li-rich
anthropogenic sources, and measured their Li contents and isotope compositions. Results show
that Li contents of all sampled anthropogenic sources are much higher than that of river water,
and their $\delta^7\text{Li}$ values are low ($< 13\text{‰}$), strongly supporting the impact of these sources on
WWTPs, Han River, as well as tap waters (see revised Table S5, and Figs. S3 & 4).

The manuscript uses population density and WW effluent in developing interpretations, which
suggests a link between population and number of WWT facilities or volume of treated WW per
unit population. That link is not mentioned in the text.

**Action:** On line 112 – 113, we were mentioning a link between population and volume of treated
WW per unit population but we have further explored this idea, as detailed in our response to the
Reviewer #2. Please see also the new Fig. S4 in the Supplementary Information.

Furthermore, although some waste Li would be “flushed” waste (and hence be part of municipal
WW treatment), Li from battery or other solid waste would likely come from significantly from
leakage from landfills. Distribution of these is also not mentioned in the text. Landfills are
mentioned (line 116), but it is implied that water from landfills would be part of the WW
treatment stream—unless Korean municipal water treatment differs from N. America, this is not
likely to be so. Some description of what the water treatment protocols are would be helpful.
How does water treatment for drinking water differ from that of wastewater (in most places these
streams are separate)?

**Response:** The new version of the manuscript details all the required information concerning the
water treatment protocols, landfill, and relationship between the river, the tap water and waste
waters. Only one landfill site (the Sudokwon Landfill) exists in the Seoul locality, at 35 km west
of Seoul. The leachate from this landfill therefore does not affect our river samples. Wastewaters
coming from the various treatment plants shown in Fig. 1 are more likely to impact the Han river
in Seoul and this is confirmed by the mixing diagram using isotopes shown in Fig. 4. We added 2
schematic diagrams showing the treatment protocols submitted by wastewaters and tap waters,
and their relationship with the river. Please see the new Figs. S1 & S2 in the Supplementary
Information.

*Although I think the study is one which, with some more data and clarification, is very valuable*
*to a variety of audiences, I think that in its current form it poses too much speculation.*

**Action:** We have significantly reinforced our interpretation by adding new – and until now
unexplored- isotope data of various Li-rich materials (see revised Table S5). We also provide
discharge rate from the various WWTP that explains well the regional variation too (see new Fig.
S4). We revised the text, Figures, Tables and the Supplementary Information. we believe the
revised manuscript is improved and our interpretation is strengthened.

*line 50: I wonder how much secular variation occurs in this system. Basically is one month's*
*worth of data a reasonable sample from which we can expect lasting interpretations to come?*

**Response:** Numerous publications, including those published in Nature, have used a case study
in order to unravel a key control. Also, global riverine solute fluxes to the ocean are upscaled
from data representing less than 30% of the total discharge to the ocean and about ~50% of
Earth's exorheic continental area. By choosing the highly populated metropolitan city of Seoul
(South Korea) and the Han River Basin, which carries small amounts of dissolved Li, and
constant and homogeneous $\delta^7\text{Li}$ values upstream, we are able to evidence the impact of
anthropogenic activities on Li levels in waters. We also provide, for the first time, Li isotope
compositions for several Li-rich materials including drugs batteries, detergent and compost.
Future studies of seasonal variations will certainly be of interest and will have to take our results
into account. In this sense, we think that our case study of the largest basin in South Korea is
pertinent.

*line 55: The data themselves are from ArcGIS? I should think the data were brought in from*
*individual sources and the manipulations were done with ArcGIS (see also Table S1).*

**Response:** The reviewer is correct.

**Action:** We addressed individual sources in the footnote of revised Table S1 as follows:
“^a The manipulations were done with ArcGIS 10 (Esri, Redlands, CA, USA). ^b Land use data
from Environmental geographic information service, EGIS (<https://egis.me.go.kr>). ^c Population
data from biz-gis (www.biz-gis.com).”

*line 74: This is unclear: shales are typically among the most Li-rich rocks.*

**Response:** We agree with the reviewer and noticed it is a typo.

**Action:** On line 67 – 68, we revised the text to read: “..., perhaps due to the occurrence of Li-
rich shales.”

**line 78:** replace “formations” with “formation”

**Response:** We thank the reviewer’s comment.

**Action:** On line 71, we changed to “formation”.

**line 105:** replace “starts to” with “approaches”

**Response:** We thank the reviewer’s comment.

**Action:** On line 112, we changed to “approaches”.

**line 140, 147:** replace “anthropic” with “anthropogenic”

**Response/Action:** We replaced all through the text.

**Fig. 2:** What is the difference between large and small symbols?

**Response:** We used large and small symbols to distinguish main river channel from each
tributary. Now we revised Fig. 2 in order to avoid the confusion.

**Action:** We added the caption as follows: “Half-transparent circles represent the tributary of each
river.”

Reviewers' comments:

Reviewer #1 (Remarks to the Author):

I have examined the revised MS and rebuttal by Choi et al. The study provides strong evidence that enhanced urban river Li levels are supplied by waste water treatment plants (WWTP). The reviewers raise an interesting debate on whether the Li in WWTP is due to Li ion battery waste or due to other sources such as Li-medication, and whether population density is the dominant control factor. The authors now provide additional evidence on the Li isotope composition of Li-medication, detergents and compost, supporting these compounds as potential Li sources to WWTP. The authors include these sources now in their conclusions, and abstract.

Regarding population density, I agree with the previous reviewers that a more important control factor may be the particular Li sources that supply waste water to WWTP. Argument as to why old, stored, and waste Li batteries would deteriorate on a timescale of decades are lacking. Given the high level of Li in medication, it may well be that WWTP Li levels are not governed by population density but by # of medicated people. This merits further investigation, but seems beyond the scope of this study, unless an efficient proxy for Li-medication use can be retrieved for Seoul. I therefore concur with the reviewers that it may be better to change the title to 'Impact of WWTP on Li content in river and tap water'.

All that said, I feel that the revised version of the MS is as strong as it can get. These are fascinating findings that will likely generate a broad interest. Once more I strongly recommend publication in NCOMMS.

Reviewer #2 (Remarks to the Author):

In general, the authors have considered reviewers comments and improved the manuscript. In particular, the new measurements on other types of anthropogenic inputs is a valuable addition. However, I still have a major concern about the correlation between population density and Li concentrations/isotopes and about the title of this manuscript "Impact of high population density on lithium content in river and tap water" which in my opinion is misleading. Put simply, correlation does not imply causation. What this study shows is that high Li content is not due to the number of people per se, but to the large Li input from lithium battery / therapeutic drug / food waste (possibly proportional to the population) combined with the inefficiency of wastewater treatment for Li-removal. 30 years ago, the population density in Seoul was also very high but there was no usage of Li-batteries (and maybe no Li-rich therapeutic drugs as well?) so the Li content in waste water and in the Han River was probably far lower than today. This example shows that high-population density is not (directly) the cause for high Li content in urban waters. The present-day high Li content in the Han river has more to do with industrial processes and urban policy instead of population density. In my opinion, the title needs to be changed to something like "Impacts of anthropogenic inputs on the lithium content of river and tap water in high-density population urban areas".

Apart from changing the title, my recommendation is that this manuscript can be published in Nature communication with only minor revision. See below some additional minor comments:

- Line 29: "Earth" scientists
- Line 64: replace "depleted with respect" by "lower relative to".
- Line 68: remove "in parallel"
- Line 90: remove "intimate"
- Line 91: Please see my comment above, there is not direct relation between Li and the number of people. I would reword "related to the number of inhabitants" to "related to increasing urban activities".
- Line 92: remove "at first". You already wrote in lines 53 to 55 that you collected and analyzed

influent and effluent wastewaters, you don't need to repeat this information here.

- Line 96: remove "are"

- Line 99: I have no idea what this sentence means, this needs to be re-written.

- Line 108-110: replace by "The relationship between Li isotope and concentration can be explained by release of isotopically light Li..."

- Line 124: replace "Combined all together" by "Altogether"

Line 154-156: this sentence needs to be re-written. You could say something like "in urban areas, Li isotopes are likely more sensitive to anthropogenic inputs rather than local weathering inputs".

Reviewer #3 (Remarks to the Author):

Review of Nature Communications manuscript by H-B Choi et al., "Impact of high population density on lithium content in river and tap water" (revised)

Review by Paul Tomascak, SUNY Oswego, 9/18/19

The authors have done a responsible job responding to the comments from my previous review, as well as those of the other reviewers and I agree with the conclusion that the increased river Li represents anthropogenic influence. I especially appreciate the incorporation of valuable new data on other potential anthropogenic pollutants—something the community has needed more of for a long time. Nevertheless I am still left with a disconnect over the cause and effect relationship suggested between population and Li in water if batteries are the primary anthropogenic Li source. My specific concerns map to place in the text noted below.

line 3: Sentences should not begin with abbreviations (spell out lithium). This should be checked throughout.

line 4: "their impacts"? The impacts of living organisms on Li levels?

line 22: "several species and human beings"? Several species of mammals?

line 37: "these effects" is ambiguous here.

line 56: delete "its"

line 57: L-SVEC, the original title, is now officially NIST RM 8545.

line 80: Why would a water regulation system (a dam) fractionate stable isotope?

line 92: Why on Fig. 3 are only five of six WWTP plants are represented?

line 117: Is water discharge rate ever quantified?

line 119-128: The manuscript also seems to confuse the point about tap water. If river water is the primary source for tap water then chemical similarity between the two is expected, and so this does not seem to need more than a sentence of coverage (not a paragraph).

line 129: More people means more cell phones. More cell phones means more cell phones being trashed. But where do those go? If they are not being sent to the landfill (unrelated to the wastewater in this study) then what? Some proportion that is not disposed of is probably accumulating in peoples' homes, but the manuscript seems to equate phones no longer in use that don't get put in the trash to some kind of instant environmental reactivity. Are people throwing used phones into the streets? I just can't connect the dots here.

Fig. 1: The key for coloring is population, though I presume this is people/km².

**Responses to the Reviewers' comments:**

Reviewer #1 (Remarks to the Author):

I have examined the revised MS and rebuttal by Choi et al. The study provides strong evidence
that enhanced urban river Li levels are supplied by waste water treatment plants (WWTP). The
reviewers raise an interesting debate on whether the Li in WWTP is due to Li ion battery waste
or due to other sources such Li-medication, and whether population density is the dominant
control factor.

The authors now provide additional evidence on the Li isotope composition of Li-medication,
detergents and compost, supporting these compounds as potential Li sources to WWTP. The
authors include these sources now in their conclusions, and abstract.

Regarding population density, I agree with the previous reviewers that a more important control
factor may be the particular Li sources that supply waste water to WWTP. Argument as to why
old, stored, and waste Li batteries would deteriorate on a timescale of decades are lacking. Given
the high level of Li in medication, it may well be that WWTP Li levels are not governed by
population density but by # of medicated people. This merits further investigation, but seems
beyond the scope of this study, unless an efficient proxy for Li-medication use can be retrieved
for Seoul. I therefore concur with the reviewers that it may be better to change the title to
'Impact of WWTP on Li content in river and tap water'.

All that said, I feel that the revised version of the MS is as strong as it can get. These are
fascinating findings that will likely generate a broad interest. Once more I strongly recommend
publication in NCOMMS.

**Response:** We thank the reviewer for his positive review and agree with his suggestion.

**Action:** Following the reviewer's suggestion as well as the one from the reviewer #2, we
changed the title as 'Anthropogenic lithium in river and tap water of high-density population
urban areas'.

Reviewer #2 (Remarks to the Author):

In general, the authors have considered reviewers comments and improved the manuscript. In
particular, the new measurements on other types of anthropogenic inputs is a valuable addition.
However, I still have a major concern about the correlation between population density and Li
concentrations/isotopes and about the title of this manuscript "Impact of high population density
on lithium content in river and tap water" which in my opinion is misleading. Put simply,
correlation does not imply causation. What this study shows is that high Li content is not due to
the number of people per se, but to the large Li input from lithium battery / therapeutic drug /
food waste (possibly proportional to the population) combined with the inefficiency of
wastewater treatment for Li-removal. 30 years ago, the population density in Seoul was also very
high but there was no usage of Li-batteries (and maybe no Li-rich therapeutic drugs as well?) so
the Li content in waste water and in the Han River was probably far lower than today. This
example shows that high-population density is not (directly) the cause for high Li content in
urban waters. The present-day high Li content in the Han river has more to do with industrial

processes and urban policy instead of population density. In my opinion, the title needs to be
changed to something like “Impacts of anthropogenic inputs on the lithium content of river and
tap water in high-density population urban areas”.

**Response and Action:** As replied to the reviewer #1, we changed the title as ‘Anthropogenic
lithium in river and tap water of high-density population urban areas’. On lines 149 – 151, we
also clarified in the text this precision that “Overall, our study shows that the large Li inputs
observed in the Han River come from LIB, therapeutic drug, and food waste, all likely
proportional to the population, combined with the inefficiency of wastewater treatment for Li-
removal.”

Apart from changing the title, my recommendation is that this manuscript can be published in
Nature communication with only minor revision. See below some additional minor comments:

- Line 29: “Earth” scientists

**Response/Action:** We changed to ‘Earth’.

- Line 64: replace “depleted with respect” by “lower relative to”.

**Response/Action:** We replaced it with ‘lower relative to’.

- Line 68: remove “in parallel”

**Response/Action:** We removed it.

- Line 90: remove “intimate”

**Response/Action:** We removed it.

- Line 91: Please see my comment above, there is not direct relation between Li and the number
of people. I would reword “related to the number of inhabitants” to “related to increasing urban
activities”.

**Response/Action:** We do not understand the difference with our statement that “anthropogenic
activities related to the number of inhabitants are responsible for the changes displayed by the
HR”. Thus, we prefer to leave it as it is.

- Line 92: remove “at first”. You already wrote in lines 53 to 55 that you collected and analyzed
influent and effluent wastewaters, you don’t need to repeat this information here.

**Response/Action:** We agree with the reviewer and removed the sentences on lines 93 – 95.

- Line 96: remove “are”

**Response/Action:** We removed it.

- Line 99: I have no idea what this sentence means, this needs to be re-written.

**Response/Action:** We clarified this sentence by revising the text to read: “Thus, any component
enriched in wastewaters can affect both the Han River and tap waters”.

- Line 108-110: replace by “The relationship between Li isotope and concentration can be
explained by release of isotopically light Li...”

**Response/Action:** We revised the text to read: “the relationship between Li concentration and its
isotopes can be explained by release of isotopically light Li from WWTP.”

- Line 124: replace “Combined all together” by “Altogether”

**Response/Action:** We replaced it with ‘Altogether’.

Line 154-156: this sentence needs to be re-written. You could say something like “in urban areas,
Li isotopes are likely more sensitive to anthropogenic inputs rather than local weathering inputs”.

**Response/Action:** We revised the text to read: “Finally, this study highlights that in urban areas,
Li isotopes are more sensitive to anthropogenic inputs rather than local weathering inputs and
therefore should be used with caution as a weathering proxy”.

Reviewer #3 (Remarks to the Author):

Review of Nature Communications manuscript by H-B Choi et al., “Impact of high population
density on lithium content in river and tap water” (revised)

Review by Paul Tomascak, SUNY Oswego, 9/18/19

The authors have done a responsible job responding to the comments from my previous review,
as well as those of the other reviewers and I agree with the conclusion that the increased river Li
represents anthropogenic influence. I especially appreciate the incorporation of valuable new
data on other potential anthropogenic pollutants—something the community has needed more of
for a long time. Nevertheless I am still left with a disconnect over the cause and effect
relationship suggested between population and Li in water if batteries are the primary
anthropogenic Li source. My specific concerns map to place in the text noted below.

line 3: Sentences should not begin with abbreviations (spell out lithium). This should be checked
throughout.

**Response/Action:** We check it thorough the text and corrected.

line 4: “their impacts”? The impacts of living organisms on Li levels?

**Response/Action:** We revised the text to read: “the impacts of anthropogenic inputs on Li levels
in the environment”.

line 22: “several species and human beings”? Several species of mammals?

**Response/Action:** We replaced ‘species’ with ‘organisms’.

line 37: “these effects” is ambiguous here.

**Response/Action:** We revised the text to read: “the effects of anthropogenic activities”.

line 56: delete “its”

**Response/Action:** We removed it.

line 57: L-SVEC, the original title, is now officially NIST RM 8545.

**Response/Action:** We replaced it with ‘NIST RM 8545’.

line 80: Why would a water regulation system (a dam) fractionate stable isotope?

**Response/Action:** A water regulation system (a dam) disturbs the natural sediment systems,
making sediments retained in the reservoir. It allows longer water-sediment interaction or deeper
riverbed erosion, causing potentially Li isotope variations (see Yang et al. JGR Earth surface,
accepted, <https://doi.org/10.1029/2019JF005078>).

line 92: Why on Fig. 3 are only five of six WWTP plants are represented?

**Response/Action:** Unfortunately, we could not collect the wastewater influent ‘GPW’ so that
there are only 5 WWTP data in Fig. 3.

line 117: Is water discharge rate ever quantified?

**Response/Action:** We used the data from the Ministry of Environment, Republic of Korea.

line 119-128: The manuscript also seems to confuse the point about tap water. If river water is
the primary source for tap water then chemical similarity between the two is expected, and so
this does not seem to need more than a sentence of coverage (not a paragraph).

**Response/Action:** We understand the reviewer’s point but, as mentioned on lines 99 – 101,
although river water is the primary source for tap water, it experiences rigorous purification
processes before river water as tap water is provided to consumer households (Fig. S2).
Therefore, although chemical and isotopic similarity between river water and tap water is
expected, rigorous purification processes could differentiate their chemical and isotopic
similarity. Because this study first demonstrates water treatment protocols are inefficient for Li,
we believe this paragraph is necessary.

line 129: More people means more cell phones. More cell phones means more cell phones being
trashed. But where do those go? If they are not being sent to the landfill (unrelated to the
wastewater in this study) then what? Some proportion that is not disposed of is probably
accumulating in peoples' homes, but the manuscript seems to equate phones no longer in use that
don't get put in the trash to some kind of instant environmental reactivity. Are people throwing
used phones into the streets? I just can't connect the dots here.

**Response:** We understand the reviewer's point. As raised by the reviewer #1, "it is an interesting
debate on whether the Li in WWTP is due to Li in ion battery waste or due to other sources such
as Li-medication, detergent and food". Indeed, we now focus less on batteries in the text and
propose various explanations to the high Li contents observed in the WWTP. Concerning the Li
contribution from the cellular phone waste, we think three different ways since only 1% of them
are recycled; 1) from illegal wastes within the agglomeration, 2) from incinerators since there are
four incineration facilities in Seoul, and 3) from numerous local industrial sites where these
batteries are constructed, transformed or used. We believe each of these possibilities would
deserve to be explored and quantified, but this is beyond the scope of this study.

**Action:** On line 130, we replaced "mostly" with "would".

Fig. 1: The key for coloring is population, though I presume this is people/km².

**Response/Action:** It is a typo. We revised Fig. 1b.

REVIEWERS' COMMENTS:

Reviewer #3 (Remarks to the Author):

I am sufficiently placated by the changes the authors have made in response to prior comments.
There is still a measure of text editing that will be needed (e.g., improper pluralizations or their lack).

**Responses to the Reviewers' comments:**

Reviewer #3 (Remarks to the Author):

I am sufficiently placated by the changes the authors have made in response to prior comments.
There is still a measure of text editing that will be needed (e.g., improper pluralizations or their
lack).

**Response:** We thank the reviewer for his comments.

**Action:** Following the reviewer's comments, we double-checked and carefully revised the text.